# Research Priorities and Trends on Bioenergy: Insights from Bibliometric Analysis

**DOI:** 10.3390/ijerph192315881

**Published:** 2022-11-29

**Authors:** Ruling Yuan, Jun Pu, Dan Wu, Qingbai Wu, Taoli Huhe, Tingzhou Lei, Yong Chen

**Affiliations:** 1College of Energy and Power Engineering, Lanzhou University of Technology, No. 287 Langongping Road, Lanzhou 730050, China; 2Changzhou Key Laboratory of Biomass Green, Safe & High Value Utilization Technology, Institute of Urban and Rural Mining, Changzhou University, No. 21 Gehu Road, Changzhou 213164, China; 3School of Environmental Science and Engineering, Changzhou University, No. 21 Gehu Road, Changzhou 213164, China; 4Guangzhou Institute of Energy Conversion, Chinese Academy of Sciences, No. 2 Nengyuan Road, Guangzhou 510640, China

**Keywords:** biomass, bioenergy, bibliometric method, social network analysis, co-occurrence analysis

## Abstract

Replacing fossil fuels with bioenergy is crucial to achieving sustainable development and carbon neutrality. To determine the priorities and developing trends of bioenergy technology, related publications from 2000 to 2020 were analyzed using bibliometric method. Results demonstrated that the number of publications on bioenergy increased rapidly since 2005, and the average growth rate from 2005 to 2011 reached a maximum of 20% per year. In terms of publication quantity, impact, and international collaboration, the USA had been leading the research of bioenergy technology, followed by China and European countries. Co-occurrence analysis using author keywords identified six clusters about this topic, which are “biodiesel and transesterification”, “biogas and anaerobic digestion”, “bioethanol and fermentation”, “bio-oil and pyrolysis”, “microalgae and lipid”, and “biohydrogen and gasification or dark fermentation”. Among the six clusters, three of them relate to liquid biofuel, attributing that the liquid products of biomass are exceptional alternatives to fossil fuels for heavy transportation and aviation. Lignocellulose and microalgae were identified as the most promising raw materials, and pretreating technologies and efficient catalysts have received special attention. The sharp increase of “pyrolysis” and “gasification” from 2011 to 2020 suggested that those technologies about thermochemical conversion have been well studied in recent years. Some new research trends, such as applying nanoparticles in transesterification, and hydrothermal liquefaction in producing bio-oil from microalgae, will get a breakthrough in the coming years.

## 1. Introduction

Greenhouse gas generated by fossil fuel combustion is one of the primary causes of climate change [1,2]. To achieve the goals of carbon-neutral and sustainable development, the world is acting to reduce fossil fuel consumption by developing renewable energy, such as solar energy, wind energy, and bioenergy [3,4]. Although traditional fossil fuels still account for 84% of global energy consumption in 2019, renewable energy already contributes more than 40% to the growth of global primary energy consumption [5]. According to the Global Renewables Outlook, the share of modern renewable energy in the final energy supply will increase to 28% by 2030, with approximately 43% consisting of bioenergy [6]. Therefore, bioenergy will play a critical role in the global energy supply system.

As a renewable alternative to fossil fuels, sustainably managed bioenergy is considered as carbon-neutral [7,8] and has become the fourth-largest resource pool of energy after oil, coal, and natural gas [9]. Bioenergy has therefore received great attention in many countries, partly because that biomass materials, such as agricultural by-products, energy crops, and the degradable parts of urban solid waste, are cost-effective and widely available in most regions of the world [10]. In practice, biomass is used as fuels directly or transformed into solid, liquid, or gaseous fuels through thermochemical or biochemical conversion technology [11]. Particularly, biomass is the sole renewable feedstock for producing liquid fuels [12], so some countries are using biomass materials to produce transport fuels. For example, America and Europe are the largest producers and users of biofuels, generating 72% and 17% of the world’s biofuels, respectively, and contributing to the decarbonization of the transportation sector [13]. According to the IPCC Special Report on Global Warming of 1.5 °C, combining bioenergy with carbon capture and storage technology will help to stabilize global carbon emissions at low levels in the future [14]. Due to the renewable and carbon-neutral characteristics of bioenergy, extensive articles on the topic have been published for developing economic and clean conversion methods.

Bibliometrics is a set of methods for determining the research priorities and trends of a specific field using mathematical and statistical methods basing on citation index databases [15,16]. Some new tools, such as the analysis of keyword co-occurrence networks, has been applied to some fields, such as power generation through waste incineration [17] and low-carbon development and transformation [18]. A keyword map network based on co-occurrence data aids in identifying major research trends and emerging research areas. With the steady increase of publications, some researchers analyzed the global structure of biomass and bioenergy research, such as the main focuses and key players in the world [19,20,21]. These previous studies are insightful; however, few studies investigate the critical topics and future directions to construct bibliometric maps for biomass and bioenergy using keyword co-occurrence analysis and related methods.

In this study, publications relating to bioenergy were comprehensively analyzed using bibliometrics. The contributions and impacts to the bioenergy field were investigated in terms of country, institution, and author. Additionally, keyword co-occurrence networks were visually analyzed to quickly identify global hotspots and development trends in bioenergy, thus providing a reference for researchers and decision-makers.

## 2. Methodology

Data for this bibliometric analysis was collected via Web of Science (WOS) and analyzed using the Intelligent Support System for Strategic Studies, provided by the Chinese Academy of Engineering and the software VOSviewer (version 1.6.15, Nees Jan van Eck and Ludo Waltman, Leiden, The Netherlands). The specific analysis methods were presented in Section 2.1 and Section 2.2.

### 2.1. Data Sources

Data acquisition was divided into three steps. The first step involved the determination of bioenergy keywords for retrieval. To ensure a high publication recall rate, the retrieval formulas included typical bioenergy keywords (such as bioenergy and biofuel) and keywords for various research directions within the field of bioenergy (such as biogas, bioethanol, and biodiesel). In the second step, bioenergy publications were collected from the WOS Core Collection database. Title retrieval was conducted for the period 2000 to 2020, and 47,292 publications were obtained. Finally, the search results were exported in plain-text format with full bibliographic information (title, abstract, author, author affiliations, source, publication year, and other information) for use in the bibliometric analysis. The search date was 24 January 2021 and the retrieval formula was as follows:

TI = (bioenergy OR biomass energy OR “bio-energy” OR biofuel * OR biomass fuel * OR “bio-fuel *” OR biomass briquette * OR biomass power generation OR biogas * OR “bio-gas *” OR biomethane * OR “bio-methane *” OR biodiesel * OR bioethanol * OR “bio-ethanol *” OR cellulosic ethanol * OR biohydrogen * OR energy algae OR energy microalgae OR energy macroalgae OR biomass electricity generation OR biooil * OR “bio-oil *” OR aviation biofuel * OR bio-aviation fuel * OR sustainable aviation fuel *)

### 2.2. Data Analysis

Bibliometrics involves evaluation and prediction scientometrics; it was originally applied in the library and information science fields and has recently been applied to other fields including studies in energy and fuels, environmental sciences, chemical engineering and mechanical engineering [22,23]. Bibliometrics can be used to obtain quantitative characteristics, such as author and country publication counts, and valuable insights into the evolution of a research field [24]. The numbers of annual publications, bioenergy development features of countries, and temporal trends of author keywords were obtained using Intelligent Support System for Strategic Studies, provided by the Chinese Academy of Engineering. The cooperation network of countries and institutions, and co-occurrence of keywords were assessed using VOSviewer software (version 1.6.15), which has been used for scientific metrological analyses in many fields and has powerful visualization functions [25]. The heatmap of keywords and countries were analyzed on a free online platform (http://www.ehbio.com/ImageGP/index.php/Home/Index/index.html (accessed on 16 March 2021). To distinguish the popular research topics in the bioenergy field using keyword co-occurrence analysis, unrelated (such as kinetics, modeling, and yield) and disturbing (such as biomass, bioenergy, and biofuel) keywords were excluded. Additionally, similar keywords (such as “bio-oil” and “biooil”, and “biodiesel production” and “biodiesel”) were merged.

## 3. Results and Discussion

### 3.1. Characteristics of Annual Publication Outputs

From 2000 to 2020, a total of 47,292 studies matched our search criteria were published. Figure 1 showed the temporal distribution of bioenergy publications. The development of global bioenergy research can be divided into three stages. During the first stage (2000 to 2004), the annual number of publications ranged from 100 to 300, representing a low output level; hence, this period was identified as the initial exploratory stage. During the second stage (2005 to 2011), the growth rate of publications increased by more than 20% per year (except 2010), with the highest growth rate of 69.8% occurring in 2007; this was the rapid development stage. During the third stage (2012 to 2020), the annual number of publications remained high. Although the growth rate of publications slowed, bioenergy remained the focus of intensive study. Many countries have incorporated bioenergy into their energy development strategies. Certain countries and regions (such as the USA, Brazil, and the European Union) have formulated bioenergy plans, and the implementation of related policies has promoted bioenergy development. It is expected that the share of primary energy supplied by modern bioenergy will increase to 23% by 2050 [6].

### 3.2. Country/Territory Characteristics

#### 3.2.1. Country/Territory Publication Distribution

From 2000 to 2020, over 100 countries/territories participated in bioenergy studies. The map in Figure 2 showed the geographical distribution of publications from these 100 countries/territories. Based on this distribution, the USA and China clearly had overwhelming advantages in bioenergy technology research, and most European countries also made great contributions.

The 20 most productive countries/territories, based on their total publications (TPs) relating to bioenergy, were shown in Table 1. Among the top 20 countries, nine are in Asia, seven are in Europe, and three are in America. The top 20 countries contributed 93.5% of the TPs. The top 20 countries each accounted for more than 1.8% of the percentage of publications (PP). The USA was the most productive country in terms of TPs and the number of citations per publication (CPP), contributing 18.1% of the total bioenergy publications. China, the second-most active country by TPs, ranked sixteenth by CPP and its research citation influence should be improved. Similarly, India and Brazil ranked third and fourth in terms of TPs and ranked relatively low in terms of CPP. In contrast, the CPP of England, Malaysia, Spain, and Sweden all exceeded 35, indicating that these countries have a strong influence.

#### 3.2.2. Country/Territory Academic Cooperation

The academic cooperation relationships among the top 46 productive countries/territories in the field of bioenergy research from 2000 to 2020, that each contributes at least 200 publications, were presented in Figure 3. The USA played a central role in bioenergy studies, having collaborated with every member of the network. The cooperation between the USA and China was particularly notable. China also played a critical role in bioenergy research and cooperated closely with countries such as Canada, Japan, and England. This is probably the reason why the USA and China were the two most productive countries. It should be noted that many European countries were involved in cooperative studies, with Germany playing a key role in collaborative relationships (as illustrated by the red cluster in Figure 3). The cooperation between countries could be strongly affected by the researchers’ study/work experience and the action of a funding agency. This information might be useful for the promotion of academic exchanges or supranational funding agreements. Therefore, effective international cooperation played a critical role in promoting the research in bioenergy.

#### 3.2.3. Bioenergy Development Features of Countries

Heatmap is often used to show the degree of correlation between a pair of objects. In this study, there were many co-occurrences among keywords and countries, as shown in Figure 4, allowing the research focus areas of each country to be identified. For readability and interpretability, the number of nodes for the keyword and country fields were limited to 20. The rapid development of bioenergy research was largely driven by these highly productive countries. As shown in Figure 4, the USA, China, and India have done more research in the bioenergy field than the other countries. These three countries also had the highest TPs. The term “biodiesel” co-occurred most frequently with China, followed by India, the USA, and Brazil, indicating that these countries have made great contributions to research on biodiesel. Biogas was the most representative topic for European countries, such as Germany, Italy, Sweden, and Denmark, with Germany contributing the most research. In Malaysia, more studies have been conducted on biodiesel than on other topics in the bioenergy field.

### 3.3. Institution Characteristics

#### 3.3.1. Institution Publication Distribution

The 10 most productive institutes in bioenergy research from 2000 to 2020, and their contributions, were listed in Table 2. Each institution published over 250 bioenergy studies. Among the top 10 institutions, three of them are in the USA, two in China, and the others belong to Malaysia, Brazil, and India, which is consistent with the country analysis results. The Chinese Academy of Sciences (CAS) was the most productive institution in this field (TPs = 885); however, this institution only ranked ninth in terms of CPP. Similarly, the University of Sao Paulo ranked third in terms of TPs, but last in terms of CPP. Although the American institutions were not as highly ranked, they had high CPP. The Indian Institutions of Technology and University Malaya ranked highly in terms of TPs and CPP, indicating their great academic influence in the field of bioenergy.

#### 3.3.2. Academic Cooperation among Institutions

The academic cooperation relationships among the top 30 productive institutions were presented in Figure 5, with each institution having more than 150 publications. The institutions cooperated more frequently within their own countries than internationally, particularly those in America, as indicated by the red cluster. Therefore, the USA played a significant role in the cooperation network of countries and institutions. For example, the National Renewable Energy Laboratory and Oak Ridge National Laboratory, and Michigan State University and University of Wisconsin in the USA had strong cooperative relationships. Institutions in Malaysia, such as University Malaya, University Sains Malaysia, and University Putra Malaysia, cooperated closely with domestic institutions, as illustrated by the blue cluster. In contrast, Chinese institutional research was highly concentrated; the CAS made great contributions to the bioenergy field and cooperated closely with research institutions both in China and abroad. Moreover, Technical University of Denmark exhibited significant international academic cooperation. Therefore, the cooperative relationship of institutions (such as CAS, Technical University of Denmark and ARS) can reflect the cooperation among countries where the institutions are located.

### 3.4. Author Publication Distribution

Table 3 listed the top 10 authors, each with over 60 publications on bioenergy. The top three authors are all from Malaysia. “Masjuki, H. H.” from University Malaya is the most productive author (TPs = 164), followed by “Kalam, M. A.” also from University Malaya (TPs = 100) and “Lee, Keat Teong” from University Sains Malaysia (TPs = 95). These authors have all studied biodiesel intensively, which may be due to the Malaysian energy policy; specifically, the biodiesel blending ratio is set to increase to 30% in Malaysia by 2025. “Angelidaki, Irini” from Technical University of Denmark (TPs = 94) primarily specializes in biogas preparation through anaerobic digestion. “Chang, Jo-Shu” from Taiwan National Cheng Kung University studies microalgae for biohydrogen. An analysis of the authors’ research direction demonstrates that most of their studies on bioenergy focused on biodiesel production, biogas generation, and biofuel production from microalgae. The CPP can be used to investigate a researcher’s influence within the relevant literature. For example, “Masjuki, H. H.” had 164 publications in the area of bioenergy with 70.4 CPP, suggesting he is a senior researcher working in relevant fields.

### 3.5. Main Journals Distribution

The publication distribution of the top 10 most productive journals, accounting for 28.2% of the TPs, was shown in Table 4. “Bioresource Technology” was the most productive journal with 2549 publications contributing to 5.4% of the TPs; followed by “Fuel” (TPs = 2027), “Biomass & Bioenergy” (TPs = 1481), “Renewable Energy” (TPs = 1289), and “Renewable & Sustainable Energy Reviews” (TPs = 1194). It can be observed that journals with impact factors (IF) > 8 accounted for 50%. Most of these journals had relatively high journal IF and h-Index, suggesting that the high impact journals highly valued bioenergy topics. The interesting finding was that these journals are in the subject areas most related to the energy and fuels.

### 3.6. Author Keyword Analysis: Identifying Prominent Research Areas

By analyzing the author keywords in the publications, we identified the prominent research areas in bioenergy technology. Figure 6 showed the top 20 keywords and their occurrence frequencies in a bubble chart, with changes in the bubble sizes indicating the trending of a popular topic. The hottest research area over the past twenty years has been biodiesel, followed by biogas, bioethanol, and bio-oil. Since 2011, 600–700 studies on “biodiesel” have been published per year, which is approximately double the occurrence frequency of “biogas” and triple that of “bioethanol”. However, the frequencies of “biogas” and “bio-oil” have changed dramatically, almost tripling in 2011–2020. Biomass conversion methods related to the research topics include transesterification, fermentation, anaerobic digestion, pyrolysis, and gasification. The frequencies of “pyrolysis” (1561) and “gasification” (894) were lower than those of “transesterification” (3882), “fermentation” (2044), and “anaerobic digestion” (1860). However, the occurrences of “pyrolysis” and “gasification” in publications increased sharply from 2011 to 2020. Thermochemical technologies (such as pyrolysis and gasification) are more suitable for treating biomass waste than biochemical technologies (such as anaerobic digestion and fermentation) [11]. Raw materials involved in bioenergy include microalgae, lignocellulosic biomass, and waste cooking oil. The frequencies of “microalgae” and “lignocellulosic biomass” have increased since 2015. Lignocellulose biomass is a promising raw material for producing biofuel; however, its high pretreatment cost limits its application in large-scale biofuel production [26].

### 3.7. Main Research Analysis of the Six Clusters

In this study, the 110 bioenergy keywords with a frequency of more than 75 were analyzed for co-occurrence, as shown in Figure 7. These important keywords were categorized into six clustering topics. The central node of cluster A is “biodiesel-transesterification”, which occupies the largest area, indicating that biodiesel is a hot topic in the bioenergy field; The central node of cluster B is “biogas-anaerobic digestion”, which pertains to the preparation of biomethane by upgrading biogas. The central node of cluster C is “bioethanol-fermentation”, which reveals that fermentation is the primary method of producing bioethanol; The central node of cluster D is “bio-oil-pyrolysis”, referring to bio-oil produced by the pyrolysis of woody biomass. The central node of cluster E is “microalgae-lipid”, microalgae are primarily used to prepare biodiesel; Cluster F is distributed across two zones, and its central nodes are “biomass hydrogen-gasification” and “dark fermentation”.

#### 3.7.1. Cluster A: Biodiesel and Transesterification

Cluster A indicates that the primary raw materials for preparing biodiesel comprise edible waste, curcas, palm, and soybean oils, and transesterification is currently the most widely used method for biodiesel preparation. A key factor influencing transesterification is the catalyst. Traditional homogeneous acid/alkali-based catalysts were widely used in biodiesel production; however, they are difficult to recover and cause environmental pollution. Heterogeneous catalysis has increased in popularity as it can overcome these disadvantages [27,28]. Heterogeneous nano-catalysts have attracted special attention in biodiesel production. Nanoparticles have highly variable surface areas that can improve their catalytic efficiency and product performance [29,30]. Enzyme catalysis transesterification has also gradually become a hot research topic. Cluster A also indicates that the influence of biodiesel mixture on engine performance and exhaust emissions has attracted increased attention. Biodiesel has a high oxygen content with no aromatic compounds, which can effectively reduce particulate matter emissions [31], but increases nitrogen oxide emissions [32]. Stability is a critical feature affecting the use of biodiesel, and antioxidants are particularly effective in improving biodiesel stability and reducing nitrogen oxide emissions [33].

#### 3.7.2. Cluster B: Biogas and Anaerobic Digestion

Biogas can easily be produced and stored. In addition to using biogas in power generation and heating, there is a trend of upgrading biogas to biomethane [34]. Cluster B reveals that the commonly raw materials for biogas production include bioenergy crops, food waste, municipal solid waste, and manure. Anaerobic co-digestion technology is the key research direction in anaerobic digestion. Single-substrate anaerobic digestion leads to nutrient imbalances [35]. However, anaerobic co-digestion balances nutrients, which can improve process stability and methane content [36]. Upgrading biogas to biomethane is one of the technologies receiving the most attention in the bioenergy sector. It is more valuable than using biogas in a heat-power co-generation system due to its higher environmental credit [37]. The key to biogas upgrading lies in removing carbon dioxide from the biogas. Carbon dioxide removal technologies can be classified into four types: adsorption, absorption, gas permeation, and low-temperature upgrading [38]. To further reduce the cost of upgrading biogas, the carbon dioxide separated from the biogas can be fully utilized in enhanced oil recovery and algae production [39], but few studies have been conducted on these topics.

#### 3.7.3. Cluster C: Bioethanol and Fermentation

Bioethanol is a sustainable resource that can be used as a mixed fuel to meet future transportation energy needs. Cluster C reveals that fermentation is the primary method of producing bioethanol, and the preparation of cellulosic ethanol (commonly known as second-generation bioethanol) from lignocellulosic biomass, such as switchgrass miscanthus, corn stover, and sugarcane bagasse, is the focus of current research. First-generation bioethanol is mainly derived from grain crops with high starch and sugar contents; however, these applications compete with food production for water and arable land. Lignocellulosic biomass is the most promising raw material for ethanol fermentation due to its abundance and low price, which can underpin large-scale cellulosic ethanol production [40]. To utilize lignocellulose for bioethanol production at the commercial scale, it is necessary to develop more effective lignocellulose pretreatment, lignocellulose hydrolysis, and simultaneous saccharification and fermentation technologies [41].

#### 3.7.4. Cluster D: Bio-Oil and Pyrolysis

Biomass can be pyrolyzed to generate bio-oil, biochar, and pyrolysis gas. Of the pyrolytic products, bio-oil is expected to become a drop-in fuel [20]. Pyrolysis methods can be primarily classified as slow, intermediate, or rapid. Cluster D indicates that studies on pyrolysis methods primarily concentrate on rapid pyrolysis. Rapid pyrolysis is one of the most promising methods as it can produce high-quality bio-oil [42]. However, bio-oil must be further upgraded to produce biofuels. Catalytic hydrodeoxygenation and hydrocracking in thermochemical treatments are the current hot research topics in this field. These processes are not cost-competitive as they require high temperature or pressure. Therefore, existing mild treatment methods should be incorporated into the commercialized thermochemical treatments for improving efficiency and reducing costs [43].

#### 3.7.5. Cluster E: Microalgae and Lipid

Microalgae biomass is a promising material for the production of biofuels such as biodiesel, bioethanol, biogas, and biohydrogen [44,45]. Cluster E indicates that microalgae are primarily used in biodiesel production. Microalgae can produce 20 times as many lipids per unit area as traditional crops [44] and require less production space. The primary hindrance in biodiesel preparation using microalgae with a high oil content is the scaling-up of production [46]. Suitable equipment/systems for large-scale cultivation must be selected. Highly productive photobioreactor cultures have recently been studied. To reduce the culture cost, microalgae cultures from wastewater have also become a hot research topic. Wastewater can be simultaneously treated and used as a source of water and nutrients for microalgae [47,48]. Microalgae biomass is diluted, and drying microalgae can account for 30% of the total production cost [49], which also poses a great challenge to the commercial-scale harvesting and extraction of microalgae [50]. Therefore, biomass conversion methods that can directly act on wet algae biomass (such as hydrothermal liquefaction) are urgently required [51].

#### 3.7.6. Cluster F: Biohydrogen and Gasification or Dark Fermentation

The only product of hydrogen combustion is water; thus, hydrogen is considered to be the cleanest energy source. Cluster F indicates that hydrogen can be produced from renewable biomass via two types of method: thermochemical and biological conversion. Biohydrogen production by gasification and steam reforming in thermochemical transformation has been a breakthrough in industrial applications. Further research of biohydrogen production through gasification is required, to increase the hydrogen yield, reduce the gasification temperature, and develop cost-effective catalysts for promoting tar cracking. Sustainable application of tar, a by-product of gasification, plays a decisive role in converting gasification into a commercially competitive biohydrogen production method [52,53]. Catalytic bio-oil steam reformation can achieve a high conversion rate; however, in practice, catalyst deactivation due to carbon deposition still hinders its technological development [54]. Additionally, dark fermentation for hydrogen is also a primary research focus in biological conversion. Dark fermentation can produce hydrogen more rapidly than other biological methods [55] and can utilize low-value organic waste, such as organic-rich wastewaters [56], combining the advantages of biofuel production with those of waste treatment.

## 4. Conclusions

Publications on bioenergy technologies from 2000 to 2020 were analyzed using bibliometric method. At global scale, the USA contributed the most publications with higher academic influence and the cooperation between the USA and China was particularly notable. Countries in European had closer cooperative relationships. The co-occurrence network of keywords indicates that liquid biofuels, including biodiesel, bioethanol, and bio-oil, are the most popular bioenergy application, attributing to liquid biofuel playing an important role in sectors that are hard to decarbonization. Raw material pretreatment and efficient catalyst preparation technologies are particularly important for converting biomass into biofuel. Additionally, upgrading biogas to biomethane is another technology receiving much attention in the bioenergy sector, as biomethane possesses the benefits of natural gas while being carbon-neutral. In conclusion, bibliometric analysis can be used to assess the progress in the bioenergy field and provide a useful reference for researchers and decision-makers. However, the title retrieval method used in this paper is not sufficient to fully reflect the subject matter of the publications. So, thematic suffix retrieval can be conducted for a more comprehensive analysis in future studies.

## Figures and Tables

**Figure 1 ijerph-19-15881-f001:**
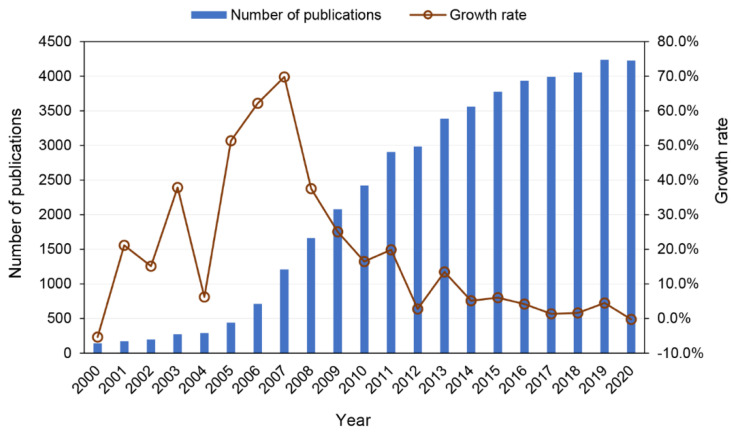
Annual number of bioenergy publications for the period 2000 to 2020.

**Figure 2 ijerph-19-15881-f002:**
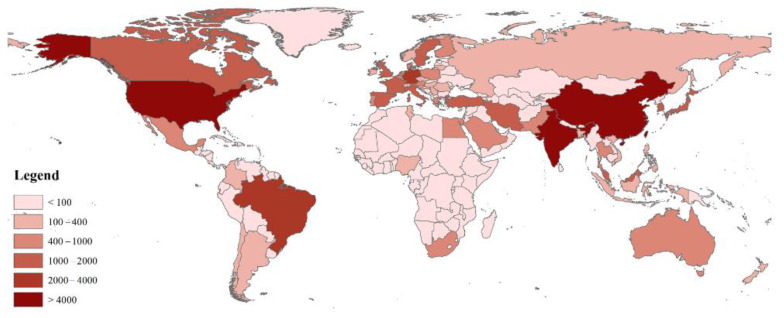
Geographical distribution of publications relating to bioenergy technology.

**Figure 3 ijerph-19-15881-f003:**
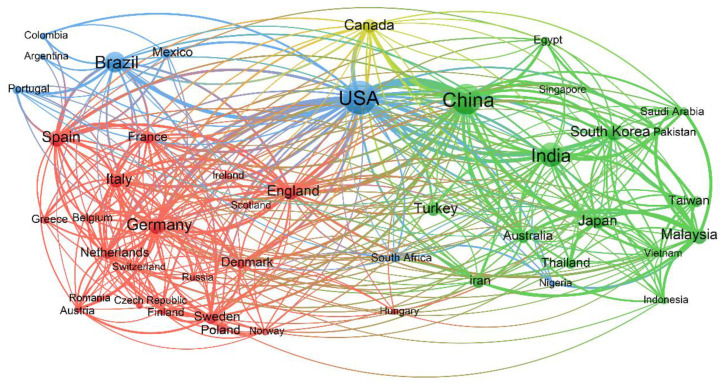
Academic cooperation network among the top 46 productive countries/territories. Larger nodes indicate more publications, and thicker lines indicate closer academic cooperation.

**Figure 4 ijerph-19-15881-f004:**
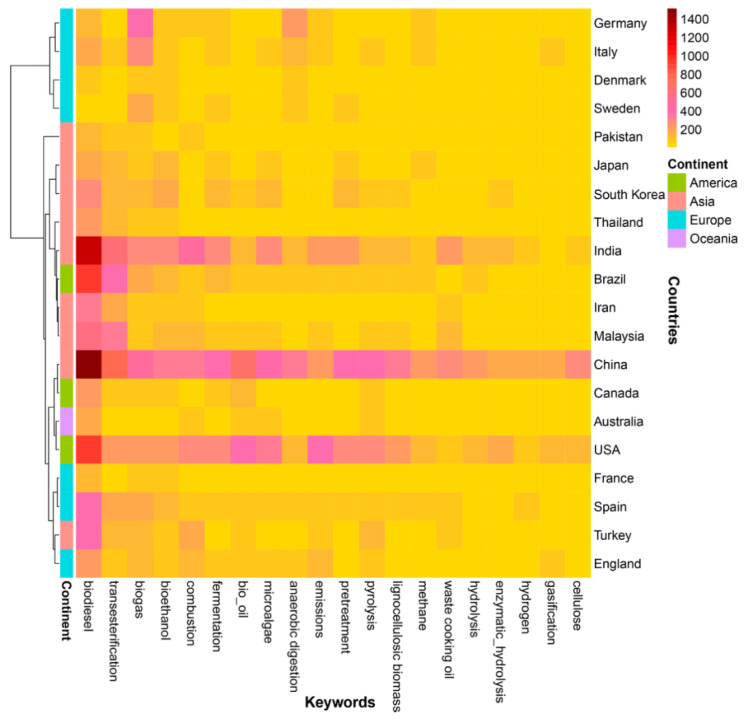
Heatmap of keywords and countries using the distance matrix method of Pearson. The color of each heatmap cell represents the number of co-occurrences for keyword and country. Unrelated keywords were excluded from the co-occurrence analysis.

**Figure 5 ijerph-19-15881-f005:**
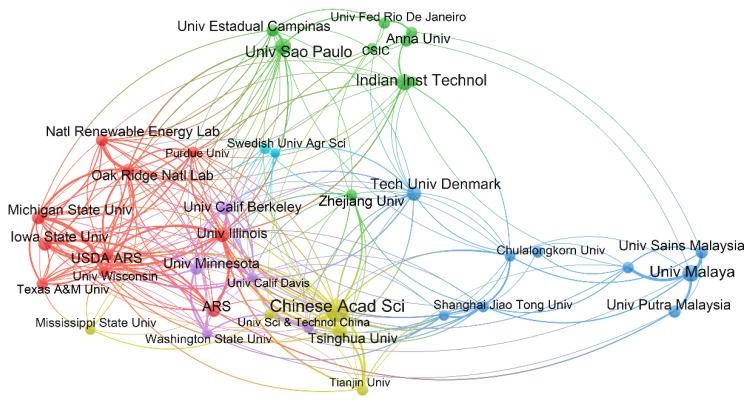
The academic cooperation network among the top 30 productive institutions. Larger nodes indicate more publications, and thicker lines indicate closer academic cooperation.

**Figure 6 ijerph-19-15881-f006:**
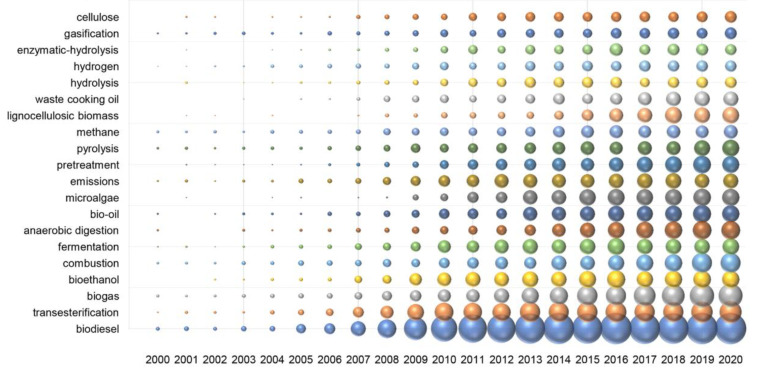
Top 20 most frequently keywords associated with bioenergy technology by year. The size of the bubble represents the occurrence frequency of the keyword.

**Figure 7 ijerph-19-15881-f007:**
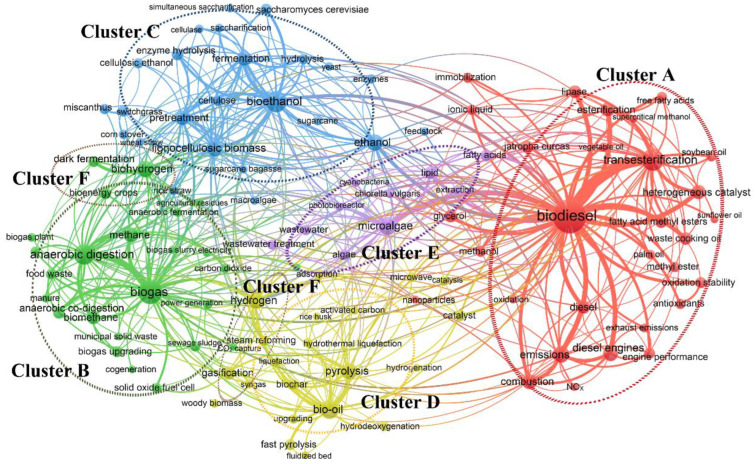
Co-occurrence network of 110 bioenergy keywords with a frequency of more than 75. Different colors areas represent different research fields on bioenergy.

**Table 1 ijerph-19-15881-t001:** Top 20 countries/territories by the total number of bioenergy publications.

No.	Country	TPs	PP (%)	TC	CPP
1	USA	8545	18.1	333,413	39.0
2	China	6917	14.6	175,891	25.4
3	India	4400	9.3	121,805	27.7
4	Brazil	3024	6.4	57,064	18.9
5	Germany	2077	4.4	54,111	26.1
6	Spain	1885	4.0	70,437	37.4
7	England	1661	3.5	64,762	39.0
8	Malaysia	1634	3.5	62,900	38.5
9	Italy	1613	3.4	44,765	27.8
10	South Korea	1555	3.3	36,034	23.2
11	Canada	1461	3.1	49,125	33.6
12	Japan	1438	3.0	42,002	29.2
13	Turkey	1212	2.6	48,115	39.7
14	France	1097	2.3	33,762	30.8
15	Sweden	1078	2.3	39,017	36.2
16	Iran	1030	2.2	20,045	19.5
17	Australia	995	2.1	31,900	32.1
18	Thailand	897	1.9	17,789	19.8
19	Taiwan	849	1.8	26,067	30.7
20	Poland	833	1.8	10,133	12.2

Legend: TPs, Total publications; PP, Percentage of the publications; TC, Total citations; CPP, Citations per publication.

**Table 2 ijerph-19-15881-t002:** Top 10 institutions by number of bioenergy publications.

No.	Institution	Country	TPs	PP (%)	TC	CPP
1	CAS	China	885	1.9	27,064	30.6
2	University Malaya	Malaysia	452	1.0	23,067	51.0
3	University of Sao Paulo	Brazil	438	0.9	9997	22.8
4	Indian Institutes of Technology	Indian	395	0.8	22,528	57.0
5	Technical University of Denmark	Denmark	355	0.8	18,085	50.9
6	Tsinghua University	China	336	0.7	12,812	38.1
7	University of Illinois	USA	334	0.7	13,498	40.4
8	Iowa State University	USA	284	0.6	13,252	46.7
9	University Putra Malaysia	Malaysia	260	0.5	8171	31.4
10	Michigan State University	USA	256	0.5	11,389	44.5

Legend: TPs, Total publications; PP, Percentage of the publications; TC, Total citations; CPP, Citation per publication.

**Table 3 ijerph-19-15881-t003:** Top 10 authors by their number of bioenergy publications.

No.	Author	Institution	TPs	TC	CPP
1	Masjuki, H. H.	University Malaya	164	11,550	70.4
2	Kalam, M. A.	University Malaya	100	5435	54.4
3	Lee, Keat Teong	University Sains Malaysia	95	5647	59.4
4	Angelidaki, Irini	Technical University of Denmark	94	5842	62.1
5	Chang, Jo-Shu	National Cheng Kung University	83	5436	65.5
6	Kumar, Gopalakrishnan	Yonsei University	80	1285	16.1
7	Ruan, Roger	University of Minnesota	79	3232	40.9
8	Ong, Hwai Chyuan	University Malaya	78	2779	35.6
9	Murphy, Jerry D.	University College Cork	69	2690	39.0
10	Rashid, Umer	University Putra Malaysia	69	2214	32.1

Legend: TPs, Total publications; TC, Total citations; CPP, Citation per publication.

**Table 4 ijerph-19-15881-t004:** Top 10 of the main journals on bioenergy.

No.	Journal Name	TPs	PP (%)	IF 2020	h-Index
1	Bioresource Technology	2549	5.4	9.642	251
2	Fuel	2027	4.3	6.609	181
3	Biomass & Bioenergy	1481	3.1	5.061	156
4	Renewable Energy	1289	2.7	8.001	157
5	Renewable & Sustainable Energy Reviews	1194	2.5	14.982	222
6	Energy & Fuels	1127	2.4	3.605	169
7	International Journal of Hydrogen Energy	1021	2.2	5.816	187
8	Energy	955	2.0	7.147	158
9	Energy Conversion and Management	865	1.8	9.709	163
10	Applied Energy	843	1.8	9.746	163

Legend: TPs, Total publications; PP, Percentage of the publications; IF 2020, the journal’s impact factor in 2020.

## Data Availability

Not applicable.

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
