# Peer review of "Research Priorities and Trends on Bioenergy: Insights from Bibliometric Analysis"

_ijerph, 2022, doi:10.3390/ijerph192315881_

Round 1

Reviewer 1 Report

The maniscript is interesting. It is suggested:

Introduction
-It is suggested to check the citatio, should be numbered in order of appearance and indicated by a numeral or numerals in square brackets—e.g., [1] or [2,3], or [4–6].

Materials and Methods
-It is suggested to place the date in which the search was carried out

Results
-It is suggested to place the top 10 of the main journals in which bioenergy topics are published
-It is suggested to place the limitations of the study

Author Response

Introduction

-It is suggested to check the citatio, should be numbered in order of appearance and indicated by a numeral or numerals in square brackets—e.g., [1] or [2,3], or [4–6].

Answer: We modified the reference format of the article as you requested.

Materials and Methods

-It is suggested to place the date in which the search was carried out

Answer: We added the search date according to your comment (2.1 Data Sources).

Results

-It is suggested to place the top 10 of the main journals in which bioenergy topics are published

Answer: According to your comment, we added the top 10 journals distribution (3.5 Main journals distribution).

-It is suggested to place the limitations of the study

Answer: We added this sentence However, the title retrieval method used in this paper is not sufficient to fully reflect the subject matter of the publications. So, thematic suffix retrieval can be conducted for a more comprehensive analysis in future studies. (4. Conclusions). This may be the limitation of the study.

Reviewer 2 Report

-        A verification should be done in the sentence –  “Masjuki H. H.” from the Universiti Malaya is the most productive author (TPs = 164), followed by “Kalam M. A.” from the Universiti Ma[1]laya (TPs = 100) and “Lee Keat Teong” from the Universiti Sains Malaysia (TPs = 95).

-        Recommendation of the name of the subtitle – “3.6. Main research analysis of the six clusters”

-         According to the instructions, the listing of references in the text must be changed.

-        After the retrieval formula, the differences between terms in quotation marks and terms without quotation marks and terms with asterisks should be explained.

-        Figure 4 should be explained in more detail. The principle of linking states is not explained.

-         It is recommended that the list of references be organised according to the journal's instructions (recommended by the ACS Style Guide).

-         The listing of references 11, 13, 16, 28 needs to be reviewed and changed.

-        The authors should interpret their results in light of previous studies and working hypotheses. The limitations of the work should be pointed out.

-         Research conclusions are recommended to be short and concise.

Author Response

- A verification should be done in the sentence –  “Masjuki H. H.” from the Universiti Malaya is the most productive author (TPs = 164), followed by “Kalam M. A.” from the Universiti Ma[1]laya (TPs = 100) and “Lee Keat Teong” from the Universiti Sains Malaysia (TPs = 95).

Answer: According to your comment, we changed the sentence “ Masjuki H. H.”, from the Universiti Malaya, is the most productive author (TPs = 164), followed by “Kalam M. A.” from the Universiti Malaya (TPs = 100) and “Lee Keat Teong” from the Universiti Sains Malaysia (TPs = 95).” to “ Masjuki, H. H.” from University Malaya is the most productive author (TPs = 164), fol-lowed by “Kalam, M. A.” also from University Malaya (TPs = 100) and “Lee, Keat Teong” from University Sains Malaysia (TPs = 95).” We also changed the “Universiti” to “university” in this article.

- Recommendation of the name of the subtitle – “3.6. Main research analysis of the six clusters”

Answer: We changed the name of the subtitle according to your comment (3.7 Main research analysis of the six clusters).

- According to the instructions, the listing of references in the text must be changed.

Answer: We modified the reference format of the article according to the instructions.

- After the retrieval formula, the differences between terms in quotation marks and terms without quotation marks and terms with asterisks should be explained.

Answer: The terms in quotation marks of the retrieval formula indicate that there are letters after them such as the plural form (biofuel and biofuels), without quotation marks indicate that there is no letter after them.

- Figure 4 should be explained in more detail. The principle of linking states is not explained.

Answer: The heatmap of keywords and countries were analyzed on a free online platform (http://www.ehbio.com/ImageGP/index.php/Home/Index/index.html), using the distance matrix method of Pearson. Therefore, we added the analysis methods and principles (2.2 Data analysis and 3.2.3 Bioenergy development features of countries)

- It is recommended that the list of references be organised according to the journal's instructions (recommended by the ACS Style Guide).

Answer: We changed list of references according to the ACS Style Guide.

- The listing of references 11, 13, 16, 28 needs to be reviewed and changed.

Answer: We reviewed these four references and changed them.

- The authors should interpret their results in light of previous studies and working hypotheses. The limitations of the work should be pointed out.

Answer: We added this sentence “However, the title retrieval method used in this paper is not sufficient to fully reflect the subject matter of the publications. So, thematic suffix retrieval can be conducted for a more comprehensive analysis in future studies.” (4. Conclusions). This may be the limitation of the study.

- Research conclusions are recommended to be short and concise.

Answer: The conclusions were refined according to your comments (4. Conclusions).

Reviewer 3 Report

Useful and well-written contribution.

The Authors may consider analyzing the following issues:

·       - A considerable part of European countries' research is carried out at ERA (European Research Area) level. This might make the breakdown into national contributions, based on the nationality of the corresponding author, less reliable

·         The cooperation between countries is measured by VOSViewer using the affiliations of the authors.  It could be strongly affected by

a) researchers' of one country having studied/worked in another (I gather this is the case for US-China cooperation)

b) the action of a Funding Agency fostering supranational cooperation (e.g. the European Framework and Life Programmes)

This information might be useful for the promotion of proactive policies of academic exchanges or supranational funding agreements. Admittedly, it would require a considerable effort probably beyond the scope of this paper.

·          Considering the 30 most productive institutions might well be offset  by a larger number of smaller institutions within a country. Indeed, the cooperation between institutions is not adequately reflected by the cooperation between the countries where the institutions are located

·        The productivity of each researcher might be affected by the publication policy adopted by each team. If not already taken care of, the results should be normalized (i.e., each paper should be divided by the number of authors)

·         - As is evident from the data collected, the above-average bioenergy  research activity with respect to other scientific sectors  is considerably correlated with its industrial/economic significance (e.g. Malaysia, Brazil, Sweden). However, Spain being the first/second country in Europe  implies additional factors playing an important role. Again, understanding the reasons for this apparent "outlier" might help direct and focus the relevant policies.

I do not recommend the Authors include additional analyses, the relevant burden being beyond the scope of this paper. However they should mention these issues in the conclusions when suggesting future research activities.
